# Fabrication of Efficient and Non-Enzymatic Electrochemical Sensors for the Detection of Sucrose

**DOI:** 10.3390/s23042008

**Published:** 2023-02-10

**Authors:** Nazia Asghar, Ghulam Mustafa, Nawishta Jabeen, Asadullah Dawood, Zeenat Jabeen, Qaiser Hameed Malik, Muhammad Asad Khan, Muhammad Usman Khan

**Affiliations:** 1Sulaiman Bin Abdullah Aba Al-Khail Centre for Interdisciplinary Research in Basic Sciences (SA-CIRBS), Faculty of Basic and Applied Sciences, International Islamic University Islamabad, Islamabad 44000, Pakistan; 2Department of Chemistry, Rawalpindi Women University, Rawalpindi 46300, Pakistan; 3Department of Chemistry, University of Okara, Okara 56300, Pakistan; 4Department of Physics, Fatima Jinnah Women University Rawalpindi, Rawalpindi 46000, Pakistan; 5Department of Physics, National Excellence Institute (University), Islamabad 04524, Pakistan; 6Department of Chemistry, University of Sargodha, Sargodha 40100, Pakistan; 7Department of Physics, COMSATS University Lahore Campus, Lahore 54000, Pakistan; 8Department of Mathematics and Physics, University of Campania “Luigi Vanvitelli”, 81100 Caserta, Italy; 9National Key Laboratory of Tunable Laser Technology, Institute of Optoelectronics, Department of Electronics Science and Technology, Harbin Institute of Technology, Harbin 150080, China

**Keywords:** sucrose, maltose, molecular imprinting, reduced graphene oxide, sensor, interdigitated electrode

## Abstract

Molecularly imprinted polymers have been used for the creation of an electrochemical sensor for the detection of sucrose, which are modified by using functionalized graphene (fG). Using AIBN as the free radical initiator and sucrose as the template, imprinted polymers are synthesized. The monomer, 4,4′-diisocyanatodiphenylmethane (DPDI), has both proton donor groups (N-H or O-H) and lone-pair donor groups (C=O). By creating H-bonds with electron donor groups (C=O), the proton donor group in this polymer may interact with the sugar molecule serving as its template. The sensor signals have improved as a result of the interaction between the monomer and the template. Thermogravimetric and differential thermal analysis (TGA/DTA) curves, scanning electron microscopy (SEM), and FT-IR spectroscopy have been employed to characterize the fabricated receptors. The fabricated sensor has exhibited a limit of detection of 16 ppb for the target analyte that is highly sensitive, linear, reversible, regenerative, and selective. Moreover, the sensor’s stability, reproducibility, and reusability have been evaluated for six months, following the device’s manufacturing, and the results revealed similar responses with the percentage error of less than 1%. Most importantly, this sensor has demonstrated a quick response time, which is very sensitive, stable, and selective.

## 1. Introduction

The most widely used sweetener, sucrose, is not only extracted mostly from sugarcane but also from fruits and vegetables. Up to 99.5% of final prepared white sugar is sucrose, which is the most crucial component in the synthesis of sugar [1,2]. Animals consume sucrose to maintain their energy levels, which affects their lipid metabolism and insulin effectiveness [3,4]. The most prevalent element in foods and beverages is sucrose [5,6,7]. Consuming too much sugar worsens human health issues [8,9]. To minimize adverse effects, it is crucial to maintain low sugar consumption [10,11]. The quantification of sucrose is performed by using traditional analytical techniques like calorimetry [12,13,14], reverse-phase high-performance liquid chromatography (RP-HPLC) [15], high-performance liquid chromatography (HPLC) [16,17,18,19], florometry [20,21], calorimetry, polarimetry, and near-infrared spectroscopy [22]. These methods are time-consuming, expensive, and complex to implement, which are all major disadvantages. Chemical sensors are devices that can detect and measure specific chemicals or groups of chemicals in a provided environment. They are used in a wide range of applications, including industrial process control, environmental monitoring, medical diagnostics, and security. Chemical sensors typically consist of a sensing-element ability, which is the part of the sensor that interacts with the target chemical, and a transducer, which converts the chemical interaction into an electrical signal that can be examined and analyzed. The sensing element can be made from a variety of materials, such as polymers, ceramics, or biomolecules, and can be designed to respond to specific chemicals or groups of chemicals [23]. Electrochemical sensors have properties that can replace the aforementioned approaches because of the drawbacks of those methods [24]. Chemical sensors are simple, inexpensive, stable, highly sensitive, and specific to the particular analyte. For online and real-time analysis, electrochemical sensors can be used [25,26].

Molecular imprinted polymers [MIPs] are promising synthetic materials with recognition sites [27,28]. These recognition binding sites are highly specific to the target analyte than to other similar molecules [29]. MIPs are essential for electrochemistry to enhance the electrode response to the selected analyte [30]. These are highly attractive materials because of their highly selective properties [31,32]. The MIPs possess greater stability, easy and fast preparation, less production cost, and generation of three-dimensional (3D) cavities of specific shape, size, and special geometry of the target molecule [33]. Shortly, in MIPs synthesis, functional monomer and cross-linker are added in a suitable solvent to establish a monomer-template complex by any physical or chemical interactions [34]. Chemically, MIPs are synthesized by bulk process where reaction occurs in a homogenous mixture [35,36]. Chemical interactions occurring between monomer and analyte may be through a non-covalent bond that can generate cavities on the bulk of the MIPs [37,38,39,40,41]. Research progress on MIP sensors for the detection of sucrose has demonstrated promising results in recent years. MIP sensors utilize the specific recognition properties of molecularly imprinted polymers to selectively detect target molecules, in this case sucrose. These sensors have been shown to possess high selectivity, sensitivity, and stability for the detection of sucrose in various sample matrices and have the potential for utilization in a variety of applications [42,43].

In this work, synthesis of artificial receptors and their composites have been optimized through molecular imprinting by functionalized graphene (fG-sucrose) as nanomaterial. A thin film of synthesized receptors and its composite has been coated onto an interdigitated electrode (IDEs). The synthesized receptors and composite have been characterized by FT-IR and SEM. The sensor performance with GO-MIP (fG-composite), MIP, and NIP has been examined by LCR meter. Furthermore, selectivity behavior has been evaluated with other saccharides.

## 2. Materials and Methods

### 2.1. Reagents, Chemicals, and Instruments

Sucrose (C_12_H_22_O_11_), maltose (C_12_H_22_O_11_), glucose (C_6_H_12_O_6_), fructose (C_6_H_12_O_6_), n-hexane (C_6_H_12_), styrene (C_6_H_8_), ethylene glycol dimethacrylate (EGDMA), 2,2′-Azo-bisisobutyronitryl (AIBN), dimethyl sulfoxide (DMSO), methanol (CH_3_OH) and ethyl alcohol (C_2_H_5_OH), acetone (C_3_H_6_O), and graphene oxide (GO) were purchased from Sigma Aldrich in the highest available purity. Characterization tools including LCR meter (IET 7600 Plus Precision LCR meter), Vortex (Toption MX-F Led Digital vortex mixer), heating water bath, and sonicator were used.

### 2.2. Synthesis of Molecular Imprinted Polymer (MIP) and GO-Composite

#### 2.2.1. Synthesis of Molecular-Imprinted Polyurethane Receptors (PU-MIP) 

The following ingredients were vortexed in 1 mL of DMSO, 0.25 g of phloroglucinol (PG), 1.83 g of bisphenol A, and 1 g of DPDI for the production of molecularly imprinted polyurethane receptors. To the above solution, 3 mg of sucrose was added, which served as a template molecule. After 35 min of heating in a water bath at 65 °C, the mixture finally transformed to a light yellowish translucent gel solution. The ratio of cross-linker to monomer was kept at 30 to 70 [44].

#### 2.2.2. Molecular-Imprinted Polymer-Functionalized Graphene Composite (GO-Sucrose) Synthesis

Using the aforementioned technique, a molecularly imprinted fG-composite was synthesized by adding 1 mg of functionalized graphene. Until the gel formation, the mixture was kept in a water bath for 35 min at 65 °C.

#### 2.2.3. Non-Imprinted Polymer (NIP) Synthesis

Without a template, the corresponding non-imprinted polymer (NIP) was prepared in the exact same way. Spin coating was used to apply 10 µL of synthesized NIP solution to interdigitated electrodes (IDEs) to create recognition layers.

#### 2.2.4. Immobilization of Artificially Designed Receptors onto the Transducing Surface (IDEs) and Measurements

The surface of the transducer was covered with thin coatings (80–100 nm thick) of synthetically created receptors using a spin-coating technique (25,000 rpm) with 15 µL of the above-mentioned solution. The thin-layer-coated devices were then treated with di-ionized H_2_O to remove the template (sucrose) from MIP and left overnight at 70 °C for layer drying and hardening. Then measurements were made using a high-precision LCR meter (7600 plus precision LCR meter) with various template concentrations, such as 0, 1, 5, 10, 20, 30, 40, and 50 ppm at 20 Hz and 1000 Hz, as shown in Figure 1.

#### 2.2.5. Characterization Methods

At room temperature, FTIR in ATR mode was used. Using an Alpha Bruker 2009 spectrophotometer, the spectra were collected throughout a wavelength range of 500–4000 cm^−1^. SEM was used to examine the surface modifications of the thin film receptors deposited onto the glass slide (JSM-7601S). A Shimadzu DTG-60 thermogravimetric and differential thermal analyzer was used for the analysis at ambient temperature to 600 °C with a heating rate of 10 °C/min, keeping the sample weight of 10 mg in flowing air.

## 3. Results and Discussion

SEM images of polyurethane-based sucrose receptors are presented in Figure 2 to study the surface morphology of the NIP, MIPs, and GO/MIPs composite. Microscopic images describe the complete embedding of functionalized graphene-composite-based receptors to MIP-based receptors, as well as the adhesion of functionalized graphene to PU/MIPs. Additionally, the surface of MIPs possesses tiny pores because the polymer swelled when the template molecule was incorporated into the polymer matrix, but the surface of NIPs completely remained smooth and nonporous. As a result, the topographies of the NIP, MIP, and GO-MIP composites differ greatly from each other, indicating that the samples have been successfully polymerized.

The ATR-FTIR spectra of the GO-composite, MIP, and NIP are presented in Figure 3 to understand the chemical compositions of the samples. The polyurethane system’s IR spectra are measured between 500 and 4000 cm^−1^. For DPDI, the C-H stretching vibration displays a peak at 2963 cm^−1^, the peaks at 1631, 1375, and 1200 cm^−1^ are allocated to the C=C, O-H, and C-O groups, respectively, and the peak at 1689 cm^−1^ is associated to the C=O group. For numerous groups, including the C-H group, the NIP, MIP, and GO-MIP spectra have persisted at peaks 2958, 2957, and 2957 cm^−1^, respectively. C-O is allocated at peak 1143 cm^−1^ and C=O associated to the peak 1716 cm^−1^ for NIP, respectively, demonstrating the presence of an EGDMA cross-linker. The presence of carboxylic acid groups has been established by the O-H bending vibration at 1375 cm^−1^. Success in polymerizing the NIP, MIP, and GO-MIP is demonstrated by the reduced intensity of C=C at peak 1631 cm^−1^. The disappearance of this peak (1631 cm^−1^) strongly suggests that the cross-linkers and monomers have formed a polymer [37]. Lower transmittance intensities of C=C stretching (1635 cm^−1^) and double-bond C=O bending (553 cm^−1^) are indicative of polymerization. While the absence of these peaks in NIP validates the removal of sugars, the typical peaks centered at 1035–1149 cm^−1^ indicate the existence of polysaccharides in MIP and GO-MIP (composite). During NIP synthesis, sugar was not added during polymerization. However, for the preparation of MIP and GO-MIP, sugar was added, allowing the polymerization under the conditions mentioned in the Materials and Methods section. Sugar (sucrose) has been removed by treating both fabricated sensors with di-ionized water with continuous stirring. As the sugar is water-soluble, it leaves the cavities complementary in size, shape, and geometry in the polymeric matrix (MIP and GO-MIP), but NIP did not possess these cavities because there was no sugar. Free radical polymerization may be the cause of the additional peaks. The stretching vibrations of sucrose’s -OH are responsible for the absorption peak at 3200–3400 cm^−1^, which supports the creation of hydrogen bonds, which is not evident in NIP. One or more weak aromatic rings were attributed to the peaks at 2800–2900 cm^−1^. By contrasting the peaks observed in MIP and GO-MIP, it has been discovered that NIP do not have them.

The thermogravimetric and differential thermal analysis (TGA/DTA) curves of (a) NIP, (b) MIP, and (c) GO-MIP are presented in Figure 4, with a temperature variation of 10 °C/min, and temperature range of 60 °C to 600 °C. The first weight loss occurred between 23.33 °C and 302.75 °C, resulting in an initial mass loss of −1.759 g. Dehydration process is associated with this temperature range, as illustrated in Figure 4a. The second stage has a temperature range of 305.10 °C to 591.69 °C with a weight loss of −4.007 mg. The subsequent breakdown of the chemical bonds in NIP is aided by this weight loss. The TGA/DTA curves of the MIP are depicted in Figure 4b with a weight loss of 3.06 g at 21.12 °C to 339.49 °C. The second weight loss of 2.515 mg has occurred between 339.47 °C and 596.74 °C. There is a −5.575 g overall weight loss in the MIP TGA curve. The temperature fluctuations in the GO-MIP TGA curves are shown in Figure 4c. In these curves, the first change occurs between 34.28 °C and 373.80 °C with a weight loss of −3.891 g, while the second change occurs between 373.80 °C and 597.58 °C with a loss of −1.298 mg. The entire weight loss of these curves, as shown in Figure 4c, is −5.189 g. These TGA curve results have shown that the MIPs, NIPs, and GO-MIPs employed for glucose sensing are extremely stable.

Utilizing the non-ionic and non-covalent interaction abilities in sensors with molecularly imprinted-based receptors coated on stretchable transducer surfaces (IDEs) with identical-size-based imprints (cavities) bind to sucrose molecules in a specific way are examined. The change in electrical characteristics on the IDE surface might be brought into account by the integration of template (sucrose) molecules from sample solutions into created cavities. The number of sucrose molecules that are integrated into the cavities have modified the conductance of MIPs’ thin layer, so conductivity is exactly proportional to the incorporated molecules on the polymer matrix surface (Figure 5). Bisphenol has been used to create sucrose-imprinted receptors based on DPDI. Through the presence of electron-rich entities on the polymer surface, the produced polymer system communicates with sucrose. From Figure 5, the sensor response is evaluated by a high-precision LCR meter after being subjected to sample solutions at various concentrations ranging from 1 to 50 ppm.

Fabricated inter-digital electrodes have been used to assess the sensitivity response of MIP and NIP (IDEs). From Figure 5, sensor behavior responds to the solution concentration in a linear fashion, i.e., response increases with increasing concentration. A thin-layer-coated IDE has been added to a solution containing 0 ppm of sucrose, and it exhibits extremely low sensor signals, showing the proposed sensor’s reversibility pattern. At 0 ppm, the observed capacitance is 0, and at 1 ppm, 5 ppm, 10 ppm, 20 ppm, 30 ppm, 40 ppm, and 50 ppm of sucrose, the sensor responses of 49 nF, 129 nF, 178 nF, 302 nF, 432 nF, 542 nF, and 679 nF have been observed, respectively, with the lowest limit of detection ~31 ppb of the fabricated sensor and newly fabricated sensor showing the linearity response of the sensor with a linear co-efficient of regression (R^2^) value = 0.99, as shown in Figure 5b. When exposed to liquids with different concentrations of sucrose, NIP shows very little sensor response. The sensor profile of sucrose molecules from 1 ppm to 50 ppm concentrations demonstrates the linearity of sensor signals as a function of concentration; this change in sensor signal is due to the change in the number of sucrose molecules incorporated in the sensor moieties. The -OH (hydroxyl) groups in sucrose may have been trapped in the molecular cavities of the polymeric matrix deposited onto the IDEs, increasing the sensor response (capacitance). Due to the existence of structurally linked cavities on the surface of MIP, the coated sensor produces a significantly stronger response as compared to NIPs. As a result, MIPs grow well-structured and geometrically acceptable cavities that permit the incorporation of template molecules only in specific locations. Figure 6 represents the incorporation and removal of analyte (sucrose) from MIP.

The morphology and structure are altered in such a way that the recognition cavities should be located at the polymer surface to improve the sensitivity and sensor performance. Only graphene is a viable option for creating the imprinted polymers that possess high mechanical characteristics and a large surface area. In the event where polymerization takes place at the surface of graphene sheets, the synthetic MIPs would have a large surface area. Molecular imprints synthesized on the surface of graphene sheets will provide complete removal of template molecules, reducing the binding time to easy accessibility to the target molecule.

Graphene oxide is two-dimensional oxygen containing a functional group with bulk molecules, which can help to attach both small and macromolecules such as polymers to its reactive oxygen functionalities. Due to the ascribed electrical and mechanical properties of graphene oxide, the sensor response of the sucrose composite (fG-MIP) has significantly improved toward its analyte. It shows a signal response of 680 nF at a concentration of 50 ppm, which brings a significant change in conductance of the DPDI-based molecular-imprinted polymer. The sensor has been exposed to various concentrations, ranging from 0 to 50 ppm, and the sensor response measured has been 94 nF, 189 nF, 269 nF, 482 nF, 683 nF, 809 nF, and 1107 nF, respectively. As the concentration of the template molecule is increased, the sensor response has risen simultaneously, demonstrating the sensor’s linearity with a linear regression constant (R^2^) of 0.99, as shown in Figure 7b, and RSD remained 1.2%, indicating that the sensor has good sensitivity. For the developed sucrose sensor, the lower limit of detection is ~16 ppm, and the upper detection point is ~600 ppm. The highly dependent relationship between sensor response and time, and the sensor response of the polymeric material, can provide the composite (fG-MIP) sucrose sensor its exceptional performance. Superior response of fG-MIP (sucrose) can be related to sucrose’s ability to bind more graphene oxide when the concentration is higher. Because MIP possesses the cavities, sieves, or certain molecular imprints are available, whereas NIP reaction is substantially less than MIP- and fG-MIP-coated IDEs. The sucrose sensor has a fairly linear response, and as analyte concentration grows, the conductance increases as well. The fabricated sensor is extremely sensitive, even at a very low concentration of sugar, i.e., 1 ppm.

When the concentration of the template is increased, the sensitivity response is strong and the conductance changes, demonstrating a linear relationship between the concentration of the template and conductance, as illustrated in Figure 8. It returns to its initial value following the template’s removal with de-ionized water, demonstrating the sensor’s perfect reversibility and reproducibility. NIP, on the other hand, exhibits no appreciable change in reaction as a result of its lack of binding capacity to analyte. The ability of the template molecule to distinguish between numerous interfering species that have a similar electro-activity to the target analyte, the same shape, and a comparable oxidizing potential is one of the most crucial parts of building an electrochemical sensor. There may be hydrogen bonds and dipole–dipole forces between the template molecules and the polymer matrix, which interact with imprinted cavities. The ability of the template molecule to distinguish between different interfering species that have the same shape, a comparable oxidizing potential, and a similar electro-activity to the target analyte is one of the most crucial components of developing an electrochemical sensor. The bulk molecular-imprinting approach may be the root of any sensor’s improved responsiveness to a specific analyte.

When compared to other polymers, the topographical investigations of the polyurethane-based sucrose sensor revealed the maximum availability of surface area for the synthesis of template-identical cavities, which have significantly improved sensor performance. In Figure 8, a bar graph comparison of NIP, MIP, and GO-MIP composites is displayed. This graph demonstrates the significant value of graphene sheets. The responsiveness of the GO-MIP-composite-based sucrose sensor is 1.9 times higher than the simple MIP-based sensor when the sensor signals from MIP and GO-MIP composites are compared.

However, sensitivity measurements by themselves do not always reveal the primary binding groups in charge of analyte–MIP interactions. However, systematic selectivity investigations using the sensors can provide additional information even from simple LCR measurements. Determination of selectivity toward compounds that are structurally similar and/or compound that essentially can be viewed as “substructures” of the sucrose molecule gain such structural information. Therefore, it is pointless to conduct experiments on potential (technological) or rival substances that might be present in a real-world matrix. An MIP-based sensor shows response of (conductance) 679 nF for sucrose, whereas maltose, fructose, glucose, and n-hexane exhibiting sensor responses of 7 nF, 6 nF, 6 nF, and 5 nF, respectively, are also observed. Furthermore, a GO-MIP-composite-based sucrose sensor has also been exposed to different concentrations of sucrose and the recorded sensor response is 1008 nF. The response of the composite-based sensor for other competing molecules is 6 nF, 5 nF, 5 nF, and 2 nF for maltose, fructose, glucose, and n-hexane, respectively. The relative standard deviation is 1.5%, which indicates that the sucrose sensor is highly selective. To achieve high sensitivity, the monomer-to-cross-linker ratio has been optimized to 30:70. A higher cross-linker ratio results in three-dimensional spaces helping to increase the selectivity toward sucrose molecules. A lower cross-linker ratio in polymer synthesis can collapse the imprinting sites on the polymer matrix.

The selectivity pattern resulting from exposing the MIP and GO-MIP (composite) to 50 ppm of maltose, glucose, fructose, and n-hexane, respectively, all of which have structural elements that are also present in sucrose, are shown in Figure 9. Selectivity is one of the fundamental features of a chemical sensor required for the detection of analyte. These molecules have structural resemblance with sucrose and the composite. Sucrose and its composite have shown high response toward the sensor against other competing species. It shows that an imprinted sensor is highly selective toward its template molecule. The selectivity coefficient for maltose is greater in comparison to glucose, which means that maltose is a stronger interfering chemical specie than glucose. The greater selectivity coefficient of maltose might be due to the resemblance of geometry and conductivity of analyte and interfering molecule because both of these are disaccharides against glucose as it is monosaccharide. In the case of n-hexane having a totally different functional group, that is why it shows the very least interference. In maltose, the same functional group is present and it has a structural resemblance with analyte molecules, i.e., sucrose. Maltose and glucose molecules might be entrapped in the selective cavities of the template, i.e., sucrose.

In contrast with MIP, a greater selectivity response has been observed in nano-composite and composite sensors. This is because of a higher surface-area-to-volume ratio. This demonstrates that the GO-MIP preparation exhibits remarkable selectivity for sucrose molecules. Due to the precise geometry, structure, orientation, and spatial arrangements that are created in composite (GO-MIP) during the synthesis condition to provide a selective cavity toward sucrose molecules, composite (GO-MIP) sensors exhibit less interference from other species in the presence of sucrose. 

Reproducibility and stability are important parameters for the application of sensors. To study the reproducibility of a GO-MIP-composite-based sensor, three different sensors have been synthesized by using the same method and stored under similar conditions of temperature and pressure for one month. After a month, these fabricated sensors were exposed to a 50 ppm concentration of sucrose and their sensor response has been recorded, as shown in Figure 10a. The above-mentioned observations indicate that GO/MIP-composite-based sensor shows only a highly negligible decrease in sensor response with a relative standard deviation (RSD) of ~0.42%. Furthermore, for stability analysis of the GO-MIP-based sucrose sensor, a sensor is designed following the same procedure of synthesis and stored for a period of six months under normal conditions. 

Sensor response has been measured on a monthly basis at a 50 ppm concentration of sucrose and an RSD value of 0.9%; it describes the GO-MIP-composite-based sucrose sensor as possessing excellent stability under the above-mentioned conditions and maintaining sensor response up to 99.5% with continuous usage, as shown in Figure 10b. The excellent reproducibility and stability were described during the sensor preparation, which not only improves the sensor response of the GO/MIP films, but also locks the size and shape of the recognition caves. To enhance the applications of the newly designed sucrose sensor, a comparison table has been added containing information from the literature review. Comparison among different types of sensors is presented in Table 1.

## 4. Conclusions

In this study, novel sucrose sensors (MIP and nano-composite) through a molecular imprinting process have been developed. A reduced-graphene composite of sucrose shows a wide linear range of detection, a lower limit of detection, exceptional sensitivity, excellent selectivity, and superior reproducibility with higher stability. The lower limit of detection for sucrose (MIP) is ≈31 ppb, whereas it is ≈16 ppb with a sucrose nano-composite (composite), which makes it an ideal candidate for the development of novel sensing devices, and a molecular imprinting strategy can be further used for fabrication of other analytes and nano-composite-based electrochemical sensors. The novelty of this work is that the fabricated sensors are very simple, easy to operate, highly sensitive, selective, and reusable. The already-available sensors are just disposable, but MIP-based sensors can be used again and again by simply treating with di-ionized water.

## Figures and Tables

**Figure 1 sensors-23-02008-f001:**
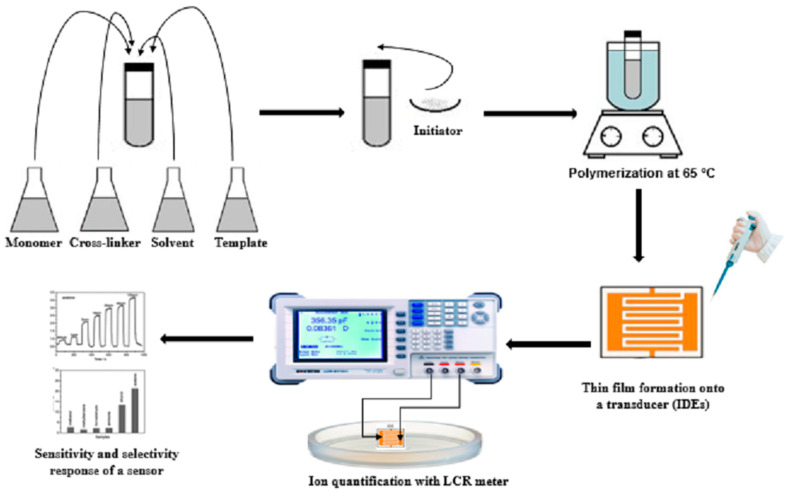
Artificial receptors synthesis and sensor setup.

**Figure 2 sensors-23-02008-f002:**
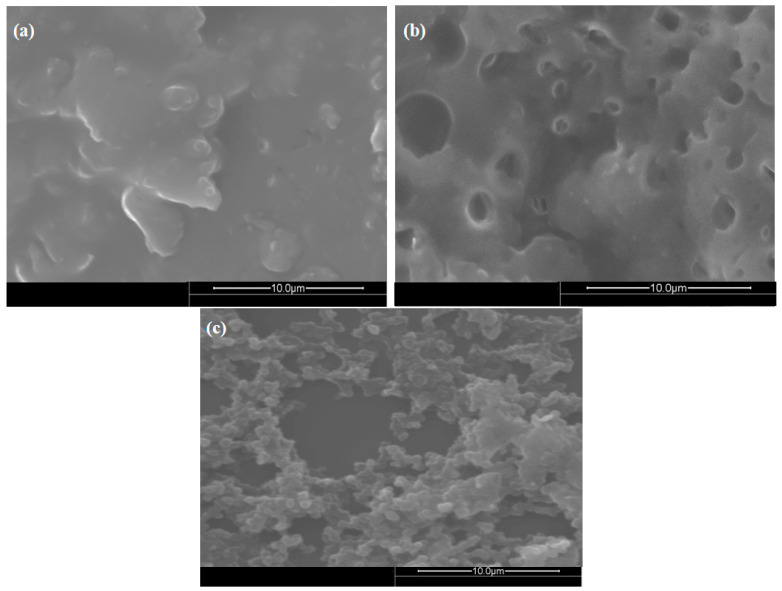
SEM images of polymer thin films coated onto transducer: (**a**) NIP, (**b**) MIP, (**c**) GO-MIP.

**Figure 3 sensors-23-02008-f003:**
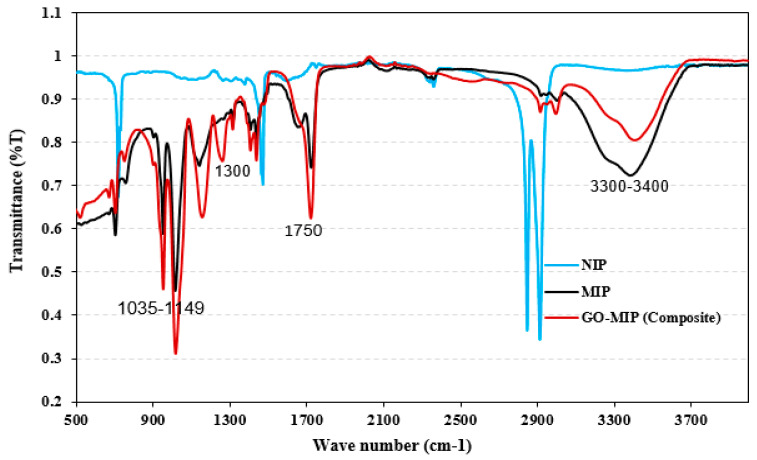
FT−IR spectra of sucrose receptors (NIP, MIP, and GO-MIP).

**Figure 4 sensors-23-02008-f004:**
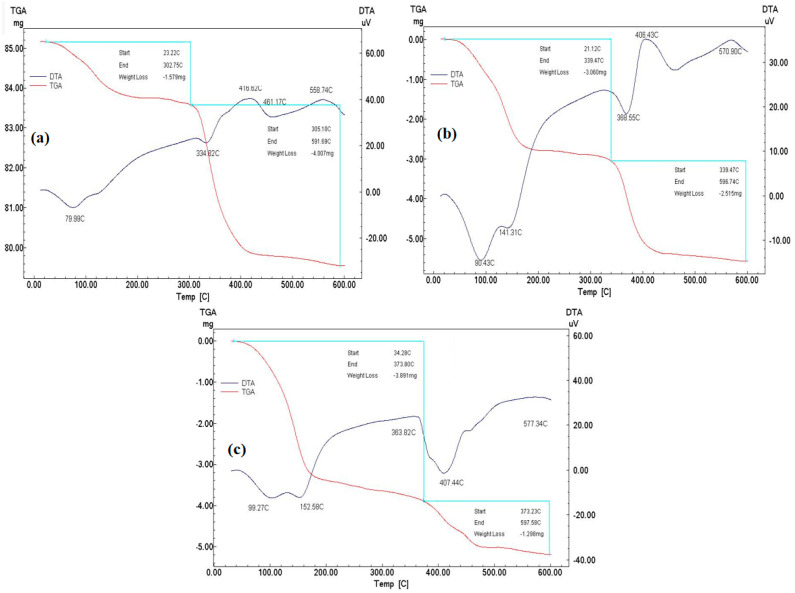
TGA and DTA curves of (**a**) NIP (**b**) MIP and (**c**) GO-MIP.

**Figure 5 sensors-23-02008-f005:**
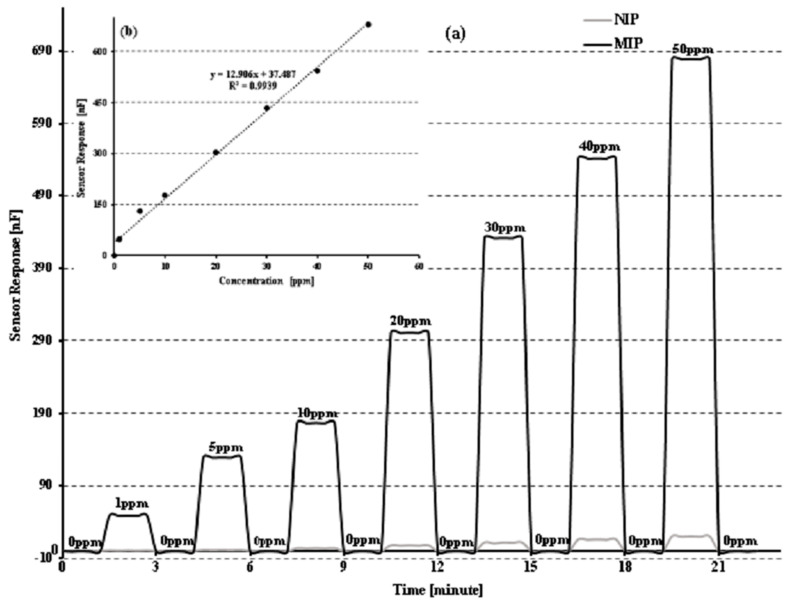
(**a**) Sensitivity response of MIPs and NIPs of polyurethane-based sucrose sensor at different concentrations (0–50 ppm); (**b**) linear regression analysis of sucrose sensor.

**Figure 6 sensors-23-02008-f006:**
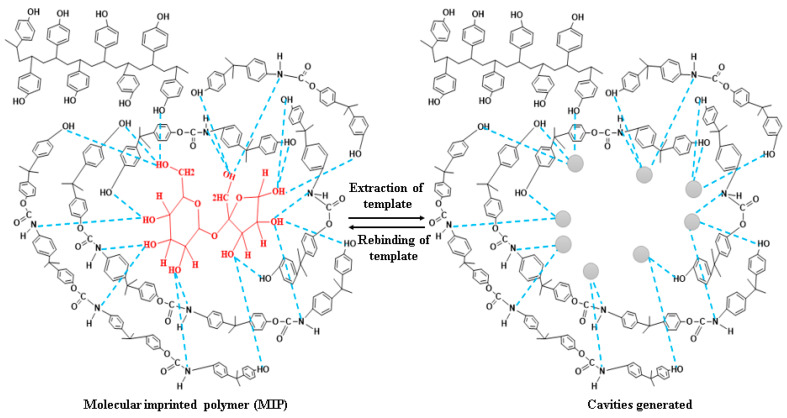
Incorporation and removal of analyte (sucrose) from MIP.

**Figure 7 sensors-23-02008-f007:**
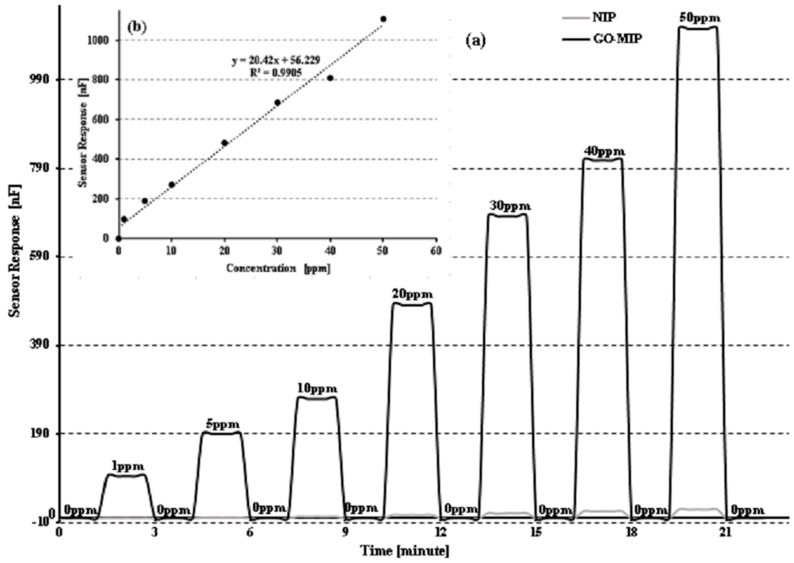
(**a**) Sensitivity response of GO-MIP composite and NIP of polyurethane-based sucrose sensor at different concentrations (0–50 ppm); (**b**) linear regression analysis of sucrose sensor.

**Figure 8 sensors-23-02008-f008:**
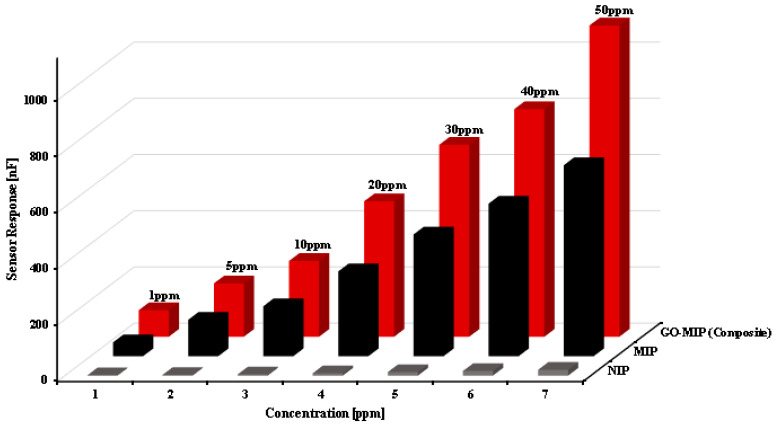
Comparison of sensitivity response of reference, MIP, and GO-MIP nanocomposite.

**Figure 9 sensors-23-02008-f009:**
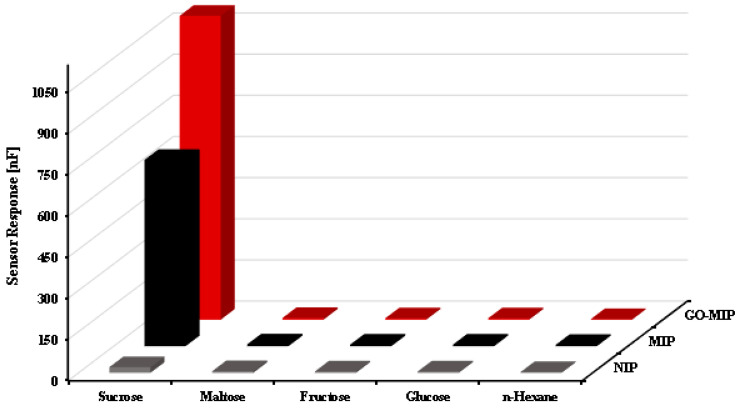
Selectivity behavior of MIP of sucrose and its composite toward maltose, glucose, fructose, and n-hexane at 50 ppm.

**Figure 10 sensors-23-02008-f010:**
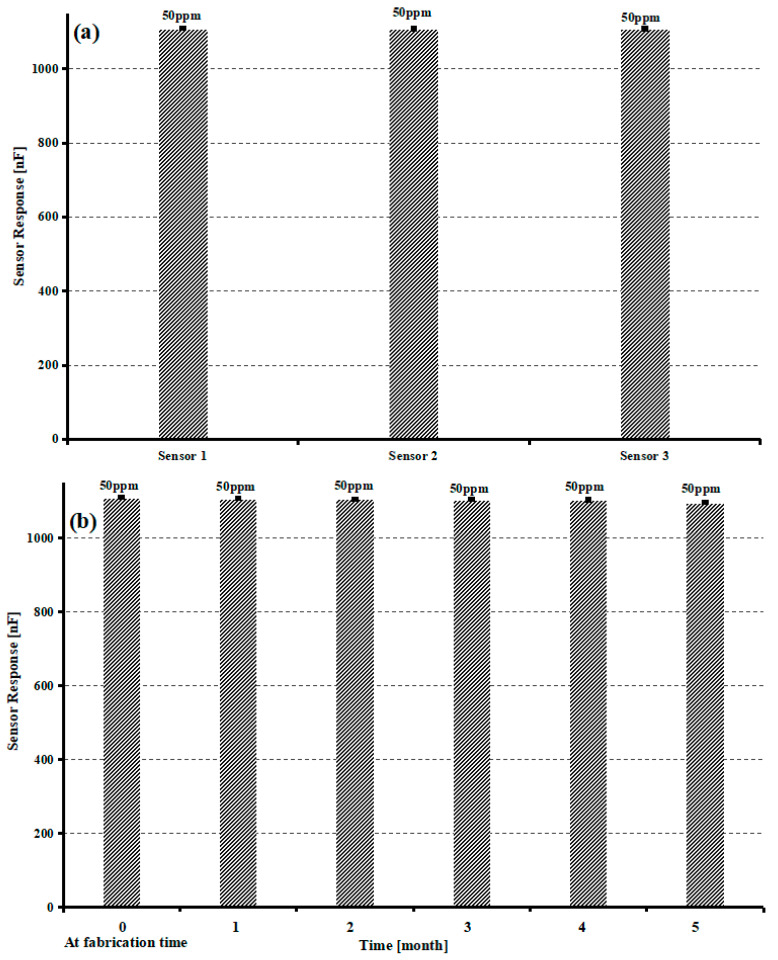
(**a**) Repeatability and stability profile of fG-sucrose (composite) (**b**) sensor over the period of 6 months.

**Table 1 sensors-23-02008-t001:** Comparison of present work with sensors cited in the literature.

Serial No.	Sensor Type	Linear Range	Lower Limit of Detection (LoD)	Reference
1	Florescent	0.025–1.0 mg/L	90,000 ppb	[45]
2	Colimetric	1 × 10^−4^–1 × 10^−9^ mol/L	1 × 10^−5^ ppb	[46]
3	Florescent	0.12–12 mg/L	1000 ppb	[47]
4	Electrochemical (MIP)	31 ppb–528 ppb	31 ppb	Present work
5	Electrochemical (Go-MIP)	16–600 ppb	16 ppb	Present work

## Data Availability

All the data is provided in this manuscript.

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
