# Peer review of "Fabrication of Efficient and Non-Enzymatic Electrochemical Sensors for the Detection of Sucrose"

_sensors, 2023, doi:10.3390/s23042008_

Round 1

Reviewer 1 Report

In the present study, the authors have attempted to synthesize a non-enzymatic sucrose-sensor on the basis of molecular imprented polymer (MIP) as well as graphene. After the successful syntheses of both the polymer and its composite and its non-imprented polymer counterpart prepared for comparison, the obtained structures were characterized by ATR-IR spectroscopy and TG-DTG analysis to determine their most possible structures. Thereafter, sensery features of the presented systems were studied. I was found that the MIP/graphene composites are very suitable in terms of selectivity, recyclability and durability.

Although both the envisioned sensor development and the collected sensor results deserved pulication, there are many weaknesses and major flaws in the presented work, which are detailed below:

1. The images/thermograms in Figures 2. and 4. are of really poor quality and difficult to understand. They should be replaced with their correct versions.

2. The language of this article is far from the level expected from a high-quality journal like Sensors. It should be improved for acceptance. In some cases, the text is almost incomprehensible.

3. No standard deviations have been added to the values exhibited, without which the explanations cannot be complete.

4. The applicability of the systems cannot be interpreted without presenting the state of the art. For this reason, the authors should add to the text a comparable table with the detection characteristics of the benchmark sensors commonly used today. It is important that the authors show both the strengths and weaknesses of their system compared to the benchmark systems.

5. In the abstract, the authors mention that the sensors have been shown to be regenerable in the prepared state, but no tests related to sensor regeneration have been presented.

6. In almost all cases, the determination of sucrose is done from blood or from its purified version or from really concentrated liquids (industry). In other words, it is no longer possible to claim that a sensor has excellent sensitivity for sucrose without performing tests in sucrose solutions with high ionic strength or at least determining the effects of the various cations and anions of the Hofmeister series. It is essential for acceptance that these tests are carried out.

7. At least one paragraph should be added to the introduction dealing with the sensors. I recommend a good example – which can also be referenced – of how this can be done: A. Szerlauth et al., Nanoclay-based sensor composites for the facile detection of molecular antioxidants, Analyst 147, 2022, 1367–1374. DOI: 10.1039/D1AN02352G.

8. In the experimental part, there is no information on sucrose detection, which makes the reproducibility of the results impossible.

Considering my above-mentioned criticisms, I recommend this article for acceptance in the MDPI journal Sensors after major revision.

Author Response

Responses for Editor and the Reviewers’

Comments

We would like to first thank the editor and the reviewer for their efforts to make this work more complete. We have carefully read all of your comments and have made necessary modifications to our revised manuscript. We believe the presentation of the paper has improved, thanks to your comments and contributions. The reviewers’ comments and our respective responses are listed below.

Reviewer 1

Comment 1. The images/thermograms in Figures 2 and 4 are of really poor quality and difficult to understand. They should be replaced with their correct versions.

Response. Thanks for the comments. We have replaced Figure 2 and 4 with their correct versions.

Actual Changes Implemented Following images have been added to the manuscript in result and discussion section at page 6.

Comment 2. The language of this article is far from the level expected from a high-quality journal like Sensors. It should be improved for acceptance. In some cases, the text is almost incomprehensible.

Response. We appreciate reviewer’s thorough review of our draft. We have revised the whole manuscript. Moreover the support from the native English speaker is also taken. I hope that the updated manuscript will fulfill the journal’s standard. Thanks for this comment.

Actual Changes Implemented According to the reviewer’s suggestions, we have improved the quality of the updated manuscript.

Comment 3. No standard deviations have been added to the values exhibited, without which the explanations cannot be complete.

Response. Thanks for this important comment. In the revised manuscript, relative standard deviations have been added at different pages.

Actual Changes Implemented, We have added the RSD at page 9, at page 11, at page 12, at page 13,  in the results and discussion section.

Comment 4. The applicability of the systems cannot be interpreted without presenting the state of the art. For this reason, the authors should add to the text a comparable table with the detection characteristics of the benchmark sensors commonly used today. It is important that the authors show both the strengths and weaknesses of their system compared to the benchmark systems.

Response Thanks for this comment. Table containing the results of fabricated sucrose sensors have been compared with findings already present in the literature.

Actual Changes Implemented. Following text and table has been added in results and discussion part of the manuscript at page 13.

Comment 5. In the abstract, the authors mention that the sensors have been shown to be regenerable in the prepared state, but no tests related to sensor regeneration have been presented.

Response Thanks for this important comment. For the purpose of re-generatability, we prepared the fabricated sucrose sensors again and again at different times following the same recipe, also measured the sensor response of these newly designed sensors and found them very closely related. Figure 10(a) showing the reproducibility and re-generatability of the sucrose sensors.

 Actual Changes Implemented Reproducibility and regeneratability is already shown in a figure 10 (a) in results and discussion part of the manuscript.

Comment 6. In almost all cases, the determination of sucrose is done from blood or from its purified version or from really concentrated liquids (industry). In other words, it is no longer possible to claim that a sensor has excellent sensitivity for sucrose without performing tests in sucrose solutions with high ionic strength or at least determining the effects of the various cations and anions of the Hofmeister series. It is essential for acceptance that these tests are carried out.

Response We highly appreciate the in-depth knowledge of the reviewer and respect his/her views. However, performing all above tests along with our reported methodology demands a complete new work. We used molecular imprinting technique (MIT) in our work where monomer, cross-linker, free radical initiator and analyte of interest, polymerized under certain conditions are used. As next step to polymerization, fabricated sensor has been treated with di-ionized water to remove analyte from the polymeric matrix. The analyte (sucrose) then leaves imprints at the surface of polymer skeleton which have been exactly identical in size, shape, geometry and functional group to the template molecule. The fabricated sensor exhibited exactly the sensitivity of template which confirms the stated results. Thus, our proposed methods do validate the claim; however, we agree that the suggestion by the reviewer would certainly make the claims more perfect. We would like to take these suggestions as an opportunity for our future work. Thanks.

Comment 7. At least one paragraph should be added to the introduction dealing with the sensors. I recommend a good example – which can also be referenced – of how this can be done: A. Szerlauth et al., Nanoclay-based sensor composites for the facile detection of molecular antioxidants, Analyst 1472022, 1367–1374. DOI: 10.1039/D1AN02352G.

Response. Thank you for the suggestion, We have added the paragraph regarding the importance of the sensors. Following text has been added and the suggested reference has also been added. Thanks for the suggestion make our work more complete.

“Chemical sensors are devices that can detect and measure specific chemicals or groups of chemicals in a given environment. They are used in a wide range of applications, including industrial process control, environmental monitoring, medical diagnostics, and security. Chemical sensors typically consist of a sensing element, which is the part of the sensor that interacts with the target chemical, and a transducer, which converts the chemical interaction into an electrical signal that can be read and analyzed. The sensing element can be made from a variety of materials, such as polymers, ceramics, or biomolecules, and can be designed to respond to specific chemicals or groups of chemicals. [A. Szerlauth et al., 2022]”

Actual Changes Implemented, we have added the following paragraph in introduction section at page 2.

Comment 8. In the experimental part, there is no information on sucrose detection, which makes the reproducibility of the results impossible.

Response. To find the reproducibility, we synthesized various sucrose sensors at different times following the same method which is already presented in material and methods. After synthesis and removal of template (sucrose) molecule, sensor response was measured time to time and found very close to each other. This confirms the reproducibility of the sucrose sensors.

Actual Changes Implemented. Text has been added at page no. 12.  

Reviewer 2 Report

Dear Editor,

The paper reports comparative work on the detection of sucrose by molecularly imprinted polymers. For sure, revealing how each step (or modification) improved the selectivity of the sensor towards the analyte is the strongest part of the paper. Besides, most portion of the results was discussed quite well. However, there some weakness in the paper must be resolved.

1- The language needs some modifications- some sentences are not clear, and word choices must be checked.

2- Figure 2 must be resized; it looks like the image was resized by only pulling one-side either down or squeezed. 

3- There are some sentences are not representing the study, for instance the sentence "The characteristics peaks centred at 1035 - 1149cm-1 indicates the presence  of polysaccharides in MIP and GO-MIP (composite) whereas absence of these peaks in  NIP confirms the removal of sugars." but during NIP preparation no sucrose was added, so how could they remove the sugar?

4- I don't think if the IR spectra were clearly annotated, details must be given..

5- I don't think if the graphene is the only promising additive for MIPs, the sentence must be revised.

6- Figure 6 depicts possible interaction between sucrose and the polymer, all relies on H- bonds; to me, C-C skeleton of sucrose can also play role during the interaction.  Of course the image is two-dimensional, so it might not be easy to reveal all the possible interactions, but still it needs to be checked.

7- There should be a table comparing the findings with the literature. 

Kind Regards,

Author Response

Responses for Editor and the Reviewers’

Comments

We would like to first thank the editor and the reviewer for their efforts to make this work more complete. We have carefully read all of your comments and have made necessary modifications to our revised manuscript. We believe the presentation of the paper has improved, thanks to your comments and contributions. The reviewers’ comments and our respective responses are listed below.

Reviewer 2

Comment 1. The language needs some modifications- some sentences are not clear, and word choices must be checked.

Response Thanks for the complete review of the submitted manuscript. We have modified the language of the whole manuscript thoroughly. Moreover the support from native English speaker is also taken in this regard.

Actual Changes Implemented. Recommended changes have been done with all the text.

Comment 2. Figure 2 must be resized; it looks like the image was resized by only pulling one-side either down or squeezed.

Response Thanks for the comment. We have replaced the Figure 2 with the original possessing good quality and high resolution.

Actual Changes Implemented. The Figure 2 of high resolution has been added in the results and discussion at page 4.

Comment 3 There are some sentences which are not representing the study, for instance the sentence "The characteristics peaks centred at 1035 - 1149cm-1 indicates the presence of polysaccharides in MIP and GO-MIP (composite) whereas absence of these peaks in NIP confirms the removal of sugars." but during NIP preparation no sucrose was added, so how could they remove the sugar?

Response We are really impressed by the understanding of this reviewer. We have rewritten the above mentioned sentence. “During non-imprinted polymer (NIP) synthesis, we didn't add the sugar during polymerization. However, for the preparation of molecular imprinted polymer (MIP) and GO-MIP, sugar was added, allowed to polymerize under mentioned conditions in material and methods section. Sugar (sucrose) has been removed by treating the both fabricated sensors with di-ionized water with continuous stirring. As the sugars are water soluble so the polymeric matrix (MIP and GO-MIP), leaves the cavities complementary in size, shape and geometry but NIP didn’t has these cavities because there was no sugar.”

Actual Changes Implemented Following changes have been added to the results and discussion part of manuscript at page no.5.

Comment 4. I don't think if the IR spectra were clearly annotated, details must be given.

Response Thanks for the comment. We have added more detail of FTIR spectra.

Actual Changes Implemented This text has been added in the results and discussion at page 5.

Comment 5. I don't think if the grapheme is the only promising additive for MIPs, the sentence must be revised.

Response. Thanks for this detailed study of the manuscript. We had revised the mentioned sentence.

Actual Changes Implemented. This text has been revised in the result and discussion section at page 8.

Comment 6. Figure 6 depicts possible interaction between sucrose and the polymer, all relies on H- bonds; to me, C-C skeleton of sucrose can also play role during the interaction. Of course the image is two-dimensional, so it might not be easy to reveal all the possible interactions, but still it needs to be checked.

Response. We are really impressed by this reviewer’s comments. You are right that C-C bond also have some interactions but our main concern is non-covalent interactions instead of covalent bonds. According to molecular imprinting, non-covalent interactions are important to remove the analyte molecule from the imprinted polymer to generate the cavities onto the polymeric matrix. That’s why we didn’t show the c-c interaction in the mentioned figure.

Comment 7. There should be a table comparing the findings with the literature.

Response Thanks for the comment. Table containing the results of fabricated sucrose sensor has been compared with findings already present in the literature.

Actual Changes Implemented Following text and table has been added in results and discussion part of the manuscript at page 13. Thank you.

Reviewer 3 Report

The paper entitled “Fabrication of efficient and non-enzymatic electrochemical sensors for the detection of sucrose” reported a novel method for detection of sucrose by using MIP sensor, which is very interesting and attractive. The idea of the paper was clearly stated, detailed studies on the preparation, detection, sensing mechanism and ability (sensitivity, excellent selectivity and superior reproducibility with higher stability) of the sensor was also presented. However, the INTRODUCTION part was not written well. 1) why the author carried out this MIP sensor work? 2) what about the research progress MIP sensors for the detection of sucrose (Kindly include advances in MIP sensors for sucrose of other researchers’ work, and relevant references should be added, such as Environmental Technology & Innovation, 2022, 28: p. 102922, Biosensors (Basel), 2022, 12(6)); 3) the novelty of this work? Furthermore, Figures' resolution should be improved (in the version I received, the resolution was not great). Correct grammar, spelling, and punctuation errors before resubmitting. So, I recommend major revision.

Author Response

Responses for Editor and the Reviewers’

Comments

We would like to first thank the editor and the reviewer for their efforts to make this work more complete. We have carefully read all of your comments and have made necessary modifications to our revised manuscript. We believe the presentation of the paper has improved, thanks to your comments and contributions. The reviewers’ comments and our respective responses are listed below.

Reviewer 3

Comment 1. Why the author carried out this MIP sensor work?

Response Because molecular imprinting is new emerging technique in which analyte molecule creates sieves/cavities exactly similar in size, shape and geometry of the template  molecule. By adsorption method, when this fabricated sensor was exposed to different concentrations of the analyte molecule, the sensor shows the response against these specific concentrations. Due to the presence of cavities onto the polymeric matrix, sensors were found highly sensitive and selective against the analyte concentration. This is because we opt to carry MIP sensors work.

Actual Changes Implemented. This text is already the part of manuscript.

Comment 2. What about the research progress MIP sensors for the detection of sucrose (Kindly include advances in MIP sensors for sucrose of other researchers’ work, and relevant references should be added, such as Environmental Technology & Innovation, 2022, 28: p. 102922, Biosensors (Basel), 2022, 12(6)).

Response. Thanks for the suggestion. We have included the suggested reference in the revised draft. More text has been added in the revised draft i.e.

“Recent research progress in MIP sensors for the detection of sucrose has shown promising results in recent years. MIP sensors utilize the specific recognition properties of molecularly imprinted polymers to selectively detect target molecules, in this case sucrose. These sensors have been shown to have high selectivity, sensitivity, and stability for the detection of sucrose in various sample matrices, and have the potential for use in a variety of applications [42-43].

Actual Changes Implemented. The text and table has been added in result and discussion section  at page 13.

Comment 3 The novelty of this work? Furthermore, Figures' resolution should be improved (in the version I received, the resolution was not great). Correct grammar, spelling, and punctuation errors before resubmitting. So, I recommend major revision.

Response We are really inspired by the technical knowledge. The novelty of this work is fabricated sensors are very simple, easy to operate, highly sensitive, selective and reusable. The already available sensors are just disposable but MIP based sensors can be used again and again by simply treating with di-ionized water. For grammer, spelling and punctuations, the complete manuscript has been revised.

Actual Changes Implemented The following changes have been added throughout the draft.

Finally, we would like to thank these reviewers for detailed technical review of our work. Their suggestions/comments were of great importance to improve the overall quality of our work. We hope the reviewers are satisfied with our revisions. Thank you again for your efforts and contributions.

Round 2

Reviewer 1 Report

The authors duly answered all my questions and adequately addressed all comments. Moreover, the changes presented make this study particularly suitable to be published. I therefore recommend the acceptance of this paper in the MDPI journal "Sensors"  in this present form.

Author Response

We research team are highly thankful to the reviewer for spending his/her precious time on our revised manuscript to make it precise and according to the standards of the highly reputed journal. 

Reviewer 3 Report

Accept.

Author Response

(The authors gave the same response as above.)
